# Design and Analysis of Ultra-Precision Smart Cutting Tool for In-Process Force Measurement and Tool Nanopositioning in Ultra-High-Precision Single-Point Diamond Turning

**DOI:** 10.3390/mi14101857

**Published:** 2023-09-28

**Authors:** Shahrokh Hatefi, Farouk Smith

**Affiliations:** Department of Mechatronics, Faculty of Engineering, the Built Environment and Technology, Nelson Mandela University, Port Elizabeth 6000, South Africa

**Keywords:** on-machine metrology, in-process metrology, hybrid machining, ultra-precision manufacturing, cutting force measurement, tool nanopositioning, diamond turning

## Abstract

Ultra-high-precision single-point diamond turning (SPDT) is the state-of-the-art machining technology for the advanced manufacturing of critical components with an optical surface finish and surface roughness down to one nanometer. One of the critical factors that directly affects the quality of the diamond-cutting process is the cutting force. Increasing the cutting force can induce tool wear, increase the cutting temperature, and amplify the positioning errors of the diamond tool caused by the applied cutting force. It is important to measure the cutting force during the SPDT process to monitor the tool wear and surface defects in real time. By measuring the cutting force in different cutting conditions, the optimum cutting parameters can be determined and the best surface accuracies with minimum surface roughness can be achieved. In this study a smart cutting tool for in-process force measurement and nanopositioning of the cutting tool for compensating the displacements of the diamond tool during the cutting process is designed and analyzed. The proposed smart cutting tool can measure applied forces to the diamond tool and correct the nanometric positioning displacements of the diamond tool in three dimensions. The proposed cutting tool is wireless and can be used in hybrid and intelligent SPDT platforms to achieve the best results in terms of optical surface finish. The simulation results are shown to be almost consistent with the results of the derived analytical model. The preliminary results pave the way for promising applications of the proposed smart cutting tool in SPDT applications in the future.

## 1. Introduction

Ultra-precision single-point diamond turning (SPDT) is a state-of-the-art technology for the advanced manufacturing of optical components [1]. SPDT technology is used to manufacture optical surfaces with an optical surface finish and an average surface roughness down to one nanometer. SPDT products have a wide range of applications in different fields of industry, aerospace, military, biomedical, electronics, and entertainment. There are different factors that influence the outcome of the SPDT process in terms of the quality of the surface finish. Different factors can negatively affect the surface generation mechanisms and increase the surface roughness [2].

It is well established that cutting force is a critical factor for the cutting mechanisms and machining conditions. Increasing the cutting force can induce tool wear and friction, decrease the diamond tool life, and negatively affect the cutting stability and machining conditions. The cutting force can also induce cutting temperature and positioning errors of the machine tool and decrease the surface accuracy of the machined product [3,4,5,6,7]. The cutting force can be monitored as a signal for monitoring tool wear and surface defects [8]. It is important to measure the cutting force during the cutting process and determine the optimum cutting parameters and machining conditions [9,10,11,12,13].

In ultra-precision SPDT applications, measuring the cutting force is important in determining the stable built-up edge and minimum chip thickness during the SPDT process. In the diamond-cutting process, the tool edge radius can be increased to compensate for the uncut chip thickness and the effect of material elastic recovery. The cutting force is used as a factor that has a correlation with the tool edge radius for modeling and estimating the minimum chip thickness. The cutting force can be used as a factor to monitor tool wear, tool life, and tool breakage [4,14,15,16,17,18,19,20,21].

To achieve the best possible machining conditions and maximum quality of surface finish with minimum surface roughness, it is important to precisely measure the cutting force in the diamond cutting of different engineering materials. Different studies have been performed to develop smart cutting tools for the in-process measurement of cutting force in SPDT processes. It is important to have a real-time measurement of cutting force with a high sampling rate, as well as a quick response of the measurement system. The measurement system should have high precision, reliability, resolution, and repeatability [13,22,23,24,25,26,27,28].

Recently, the application of hybrid machining platforms for the ultra-high-precision SPDT of different engineering materials has emerged. To enable an ideal hybrid machining platform for SPDT applications, it is necessary to use a high-performance force measurement system with a high frequency and quick response for the in-process metrology of applied force with high precision, resolution, and repeatability. In a hybrid SPDT platform, the force measurement system should be capable of real-time sampling while communicating and transmitting the acquired data to the hybrid controller for process optimization [13,29,30,31].

The purpose of this research is the design and analysis of a high-performance, wireless, smart cutting tool system for the in-process metrology of cutting forces, as well as tool position control during the SPDT process. The proposed smart cutting tool can measure the applied forces in three dimensions (*X*-, *Y*-, and *Z*-axes) in the diamond-cutting process with high precision. It can also monitor the nanometric displacements of the diamond tool and correct the position error along the *X*-, *Y*-, and *Z*-axes and correct position errors. The designed smart cutting tool can work stand-alone or as a part of a hybrid SPDT platform. In a hybrid SPDT platform, the proposed system can communicate with the hybrid controller and transmit the acquired data, including the measured cutting forces, using wireless solutions.

## 2. Background Overview

### 2.1. Smart Cutting Tools for In-Process Force Measurement in Ultra-Precision SPDT

Different techniques have been used in the development of metrology systems for the in-process measurement of cutting forces in high-precision machining applications, including piezoelectric, capacitive, optoelectronic, and strain gauge [25,32,33,34]. There have been many developments in the application of force sensors in different machining applications, including milling and lathe machining processes. However, a limited number of studies have focused on the development of specific high-precision force measurement systems with high performance for SPDT applications. In an ultra-high-precision SPDT process, the application of smart cutting tools has been emerging for improving the machining conditions and enabling the in-process metrology of machining factors, including the cutting force and cutting temperature. Figure 1 illustrates the classification of the smart cutting tools and force measurement techniques in ultra-precision SPDT applications. Two types of force measurement techniques can be used for the in-process metrology of the cutting force in SPDT processes. Acoustic waves can be used for measuring the change in the surface strain using contactless passive acoustic waves. Piezoelectric ceramic films are a category of solutions that have been widely used for different force measurement applications, including cutting force measurement in ultra-precision machining applications [28].

In 2014, Chen et al. [23] developed a smart cutting tool for the measurement of the feed force and cutting force during ultra-precision cutting. In the developed smart cutting tool, piezoelectric films were used for measuring the applied force in the SPDT process. Variation in the applied force changes the resistivity of the piezoelectric films, and the applied force can be measured by sensing the change in the resistance of the piezoelectric element. The developed technique could enable measuring the cutting force during the ultra-precision cutting process with a 0.1 N resolution within the range of 10 N. In another study by Xiao et al. [24], a smart cutting tool was developed using PZT5H piezoceramic material for the real-time measurement of cutting forces along the *X-*, *Y*-, and *Z*-axes. In this system, piezoelectric sensor arrays were implemented in the tool shank for measuring the applied force. In this system, by using a decoupling algorithm, the changes in the signals generated by the piezoelectric array were distinguished, and the cutting forces were measured in real time with a resolution of 0.1 N.

In 2016, Liang et al. proposed a sensor system for the in-process metrology of cutting forces during the cutting process. In the proposed system, six force sensors using strain gauges were used to measure the cutting force and cutting moment along the *X*-, *Y*-, and *Z*-axes [25]. In 2017, Zhao et al. [26] developed a smart cutting tool based on a strain gauge. In this smart cutting tool, three pairs of strain gauges were bonded on the tool shank, while three Wheatstone bridges were used to distinguish the changes in the resistance of the strain gauge and to measure the applied cutting force. In 2022, Odedeyi et al. [35] developed a self-sensing tool holder for in-process cutting force measurement in the SPDT process. In the developed device, a pair of strain gauges was used to measure the cutting force during the diamond-cutting process with a high sensitivity of 4.592 mV/N within the range of 3 N. By sensing the change in the resistance of the strain gauge, the tension of the surface can be calculated, and the applied cutting force can be determined accordingly.

In SPDT, small cutting forces are executed during the diamond-cutting process. Different methods have been used to enable a high-precision force measurement during the ultra-high-precision SPDT process. Although in the existing method the magnitude of the applied cutting force can be measured precisely, the positioning error of the cutting tool caused by tool holder surface deformation is not monitored or compensated for. When a strain gauge is used, the change in the surface strain determines the magnitude of the applied force. When piezoelectric components are used, the change in the physical properties of the piezoelectric material under the applied cutting force changes the electrical properties of the material, which determines the magnitude of the applied force. In both solutions, the cutting force is measured; however, the nanopositioning errors caused by the applied force are not measured or corrected. There is still a gap between the existing methods and an ideal solution to be used in intelligent and hybrid SPDT platforms.

### 2.2. Piezoelectric Actuators and Ultra-Precision Nanopositioning Systems

Piezoelectric materials and electrostrictive mechanisms are key elements in high-precision positioning systems. The main mechanical characteristics for a dynamometer are high stiffness, refit capability, rigidity, accuracy, and sensitivity. The application of piezo thin films and strain gauges in the development of precision force sensors is well established. By the development of new piezoelectric materials, the application of advanced ceramics in piezo applications, measurement systems, and high-precision actuators for nanopositioning applications has emerged [36]. Recently, piezoceramic materials have been widely used in high-precision sensor/actuator applications. Ceramics exhibiting piezoelectric properties belong to ferroelectric materials. Piezoceramics are high-performance materials for piezo applications. In the development of sensors and actuators, piezoceramic materials with a permittivity between 1000–3000, a mechanical quality factor < 100, and a high piezoelectric activity *d*_33_ ≥ 400 × 10^−12^ are usually used. SONOX and PZT materials, including PZT5H1, PZT5A4, PZT5A1, PZT503, SONOX^®^ P5, and SONOX^®^ P504, are from commonly used piezoceramic materials in high-precision sensor/actuator applications [36,37,38].

Piezoelectric materials can convert mechanical parameters such as pressure and acceleration into electrical quantities and vice versa. Piezoelectric components are stable, durable, and an ideal choice for different high-precision applications, including advanced manufacturing, automotive, defense, security, energy, medical, and industrial applications. Low-voltage piezoelectric ceramics have a quick response time and sub-micron resolution. Piezoelectric ceramics can be customized based on the required application. Piezoceramic chips are fabricated from layered sheets of piezoelectric material. Piezoelectric actuators are characterized by high-precision movement. Multi-layer piezoelectric actuators were recently used in the development of high-precision linear and non-linear positioning systems. The performance and accuracy of the piezoceramic chips as actuators rely on the capability of the piezoelectric material to produce a controlled deformation in the micrometer to nanometer range [36,37,38]. In recent studies, multi-layer piezoceramic chips/piezoelectric stack actuators were used for the development of high-precision nanopositioning systems. These components can execute displacement and force by the inverse piezoelectric effect. Multi-layer piezoceramic chips are composed of many thin layers of piezoceramic plates, electrically bonded in parallel and mechanically connected in series. This configuration enables the expansion of the thickness of the piezoceramic chips that can be precisely controlled by the applied voltage to the electrodes [39,40].

In 2012, Panda et al. [41] proposed a low-voltage amplified piezo actuator that could execute displacement at the nanometer scale. In this research, six multi-layer piezoelectric stacks were used in a diamond-shaped configuration, which could execute a maximum displacement of 173 μm at 175 VDC, with an average displacement of 988 nm/V. In 2017, Yong et al. [42] proposed a nanopositioning system by using piezoelectric stack actuators with an integrated force sensor. The proposed method could be used for the damping and tracking control of a high-speed nanopositioning stage. These studies show the application of piezoelectric actuators in the development of high-precision nanopositioning systems with a resolution at the nanometer scale. Preloaded piezoelectric stack actuators have high performance and are suitable for nanopositioning systems, as these systems can execute displacements with sub-nanometer resolution [43].

## 3. Material and Methods

### 3.1. Theory and Principles

Figure 2 illustrates the working principles and the configuration of the proposed smart cutting tool. In the design of the smart cutting tool, three preloaded piezoceramic actuators are installed on the tool holder structure in the *X*-, *Y*-*,* and *Z*-directions. This configuration enables the nanopositioning of the cutting tool to compensate for the displacements and positioning errors of the diamond tool with nanometer accuracy. One piezo stack actuator (PSA) can move the tool holder and diamond tool along the *X*-axis, in the opposite direction of the back force, and the other PSA can move them along the *Y*-axis, in the opposite direction of the applied cutting force. Another PSA can move the tool holder and diamond tool along the *Z*-axis, in the opposite direction of the feed force. By applying a controlled DC voltage to each PSA, their length can be changed in a fully controlled manner. Therefore, by applying a suitable amount of voltage to each PSA, the required change in the length of the PSA can be executed and the unwanted displacements of the diamond tool, caused by applied forces during cutting, can be compensated for.

Moreover, when an external force is applied to the PSA, it causes the generation of an electrical charge in the piezoceramic material and influences the electrical charge and electrical capacity of the PSA, which can be acquired and implemented as a factor for measuring the magnitude of the applied force in each respective direction. By monitoring the output voltage, the PSA aligned with the *X*-axis can measure the back force applied to the diamond tool, the PSA aligned with the *Y*-axis can measure the cutting force applied to the diamond tool, and the PSA aligned with the *Z*-axis can measure the feed force applied to the diamond tool. Two conductive plates are fixed to the sides of each PSA, as electrodes, for driving the component, as well as measuring the change in the electrical charge of each component, which is caused by the applied forces. A high-performance control system with a high-precision amplifier and analog-to-digital converter is designed to enable measuring of the small amounts of changes in the electrical charge generated by the PSAs and strain gauges.

The applied cutting force during machining causes deflection on the tool holder surface at the nanometer level, which causes nanometric displacements at the tool position during the cutting process. The interactive forces in diamond cutting and their directions are illustrated in Figure 2B. In force measurement applications, strain gauges can be used to measure the strain on a surface. The structure of the strain gauge is sensitive to dimensional changes when an external force is applied to the surface in which the strain gauge is bonded to. The output voltage of a Wheatstone bridge is directly proportional to bridge excitation and very sensitive to the small mechanical deformations that drive resistance imbalances in one or more legs of a bridge. For measuring the surface deflections, as well as tool position control, twelve strain gauges are bonded on three surfaces of the tool holder to precisely measure the deflections on the tool holder surfaces caused by the applied forces. The change in surface strain can be used as feedback for correcting the displacement of the diamond tool along the *X*-, *Y*-, and *Z*-axes.

#### 3.1.1. Characteristics of the Piezoelectric Component and Multi-Layer Piezo Stack Actuator

In the proposed smart cutting tool, three low-voltage PSAs are used for measuring the forces applied to the cutting tool during the diamond-cutting process, as well as correcting the positioning errors caused by the applied forces during the manufacturing process. An important aspect in the design of a high-precision force measurement system using PSAs is the material characteristics of the piezoelectric chip. When a DC voltage resource is connected to the electrodes of a piezoelectric component, its dimensions change based on the amplitude of the applied voltage, as illustrated in Figure 3A. The domain of the displacements is correlated with the amount of the applied voltage. The electrical behavior of the piezoelectric component is equivalent to the circuit diagram illustrated in Figure 3B.

The mechanical quality factor *Q_m_* is a dimensionless quality factor for calculating the mechanical losses in the dynamic performance of piezoelectric components. *Q_m_* is the ratio of input energy per oscillation cycle over the total energy consumed in the cycle. This factor can be calculated as follows:(1)$$Q_{m}=\frac{f_{p}^{2}}{2\cdot \pi \cdot f_{s}\cdot Z_{s}\cdot C\cdot (f_{p}^{2}-f_{s}^{2})}$$
where *Q_m_* is the mechanical quality factor, *f_p_* is the parallel resonance frequency, *f_s_* is the series resonance frequency, *C* is capacitance, and *Z_s_* is the impedance at the resonance frequency. The dielectric relative permittivity of a piezoelectric component *ε**_r_* is defined as the ratio between the absolute dielectric constant and the permittivity of free space *ε*_0_ = 8.85 × 10^−12^ F/m. The capacitance of a piezoelectric component is measured at a frequency below the resonance frequency of the piezoceramic, usually at 1 KHz. The capacitance can be calculated using the following equation:(2)$$C=\varepsilon_{r}^{T}\cdot \varepsilon_{0}\cdot (\frac{A}{D})$$
where *C* is the capacitance, *ε**_r_* is the relative dielectric permittivity, *ε*_0_ is dielectric permittivity of free space, *A* is the area of the electrode, and *D* is the electrode spacing. The capacitance of a rectangular plate piezoceramic can be simplified to the following equation:(3)$$C=\frac{\varepsilon_{r}\cdot L\cdot W}{T\cdot 113}\;[pF]$$
where *C* is the capacitance, ε is the relative dielectric permittivity, *L* is the length, *W* is the width, and *T* is the thickness of the piezoceramic material.

When a range of different frequencies is applied to the piezoceramic component, the series resonance frequency (*f_s_*) can be obtained at the minimum impedance, and the parallel resonance frequency (*f_p_*) can be obtained at the maximum impedance. The resonance frequency of a plate piezoceramic can be calculated using the following equation:(4)$$f_{r}=\frac{N_{1}}{L\;or\;W}$$
(5)$$f_{r}=\frac{N_{T}}{T}$$
where *N* is the frequency constant of the longitudinal/transverse oscillation of the piezoceramic plate, *N_T_* is the frequency constant for the thickness mode oscillation of the piezoceramic plate, *L* is its length, *W* its width, and *T* its thickness.

The piezoelectric charge constant (*d*) denotes the rate of generated electrical charge and the applied force. The charge constant of piezoelectric components can be measured using the following equation:(6)Q=d·F
where *Q* is the generated electrical charge, *d* is the charge constant, and *F* is the applied force. The voltage and charge constants denote the ratio between the force applied and the amount of electric field generated. These two constants are in correlation with the capacitance of the piezoceramic component. Therefore, by measuring the generated electrical charge, the magnitude of the applied force can be determined. The charge constant of a piezoelectric component is expressed by the following equation:(7)$$d=\varepsilon_{0}\cdot \varepsilon_{r}\cdot g$$
where *d* is the piezoelectric charge constant, *ε_r_* is the relative dielectric permittivity, *ε*_0_ is dielectric permittivity of free space, and *g* is the piezoceramic voltage constant.

The above characteristics of the piezoelectric materials are used for determining the characteristics of the multi-layer piezo stack actuators. In the design of the smart cutting tool, three PSAs are used. A multi-layer piezoelectric actuator is produced based on a unique element structure design and piezoelectric ceramic materials with high electro strictive factors. The piezoceramic chip consists of stacked piezoelectric ceramic layers between the integrated electrodes. These ceramic layers are mechanically in series and the electrodes are electrically parallel. The capacitance of a rectangular plate piezo stack can be calculated using the following equation:(8)$$C=\frac{\varepsilon_{33}^{T}\cdot L\cdot W}{T}$$
where *C* is the capacitance of the piezoceramic component, ${\varepsilon \;}_{33}^{T}$ is the relative dielectric permittivity in the 33 polarization direction, *L* is the length, *W* is the width, and *T* is the thickness of the piezoceramic component. In this condition, the executed length, transverse, and thickness displacements can be calculated using the following equations:(9)$$∆W=\frac{d_{31}\cdot W\cdot V}{T}$$
(10)$$∆L=\frac{d_{13}\cdot L\cdot V}{T}$$
(11)$$∆T=d_{33}\cdot V$$
where ∆*W* is the width displacement, ∆*L* is length displacement, and ∆*T* is thickness displacement. *d*_31_ is the piezoceramic charge constant in the 31 polarization direction, *W* is the width, *V* is the electrical voltage applied, *L* is the length, and *T* is the thickness of the piezoelectric component. *d*_13_ is the piezoceramic charge constant in the 13 polarization direction and *d*_33_ is the piezoceramic charge constant in the 33 polarization direction. A specific amount of voltage should be applied to the piezoceramic component to cause the desired displacements and to achieve the position correction in each axis (*X*, *Y*, *Z*), as illustrated in Figure 2. The amount of the required voltage can be calculated using the following equations:(12)$$V=\frac{g_{31}\cdot F_{1}}{W}=\frac{g_{13}\cdot {\;F}_{2}}{L}=\frac{g_{33}\cdot F_{3}\cdot t}{L\cdot W}$$
where g is the voltage constant, *F* is the generated/applied force, *t* is the thickness, *W* is the width, and *L* is the length of the piezo stack. In the designed system, the applied force can be calculated by measuring the electrical voltage generated across the piezoceramic component by means of the voltage constant, using following equation:(13)$$U=g_{33}\cdot h\cdot \frac{F}{A}$$
where $g_{33}$ is the voltage constant of the piezoceramic in the 33 polarization direction, *h* is the height of piezoceramic element, *F* is the applied force, and *A* is the area of piezoceramic component. Figure 3 illustrates a piezoceramic rectangular plate and its application in executing vibration and displacement.

Figure 4 illustrates the typical performance of the PSA. It can be seen in the graph of Figure 4A that by applying a variable DC voltage to the PSA, different amounts of displacements can be achieved. Therefore, by controlling the applied voltage at a micro-volt precision, executing displacements with nanometer precision can be achieved. The designed closed-loop control system can also compensate for the hysteresis position errors by changing the amount of the applied voltage to the PSAs. Figure 4B illustrates the frequency vs. temperature graph of the PSAs. The natural frequency of the designed system should be four times higher than the natural frequency of the SPDT machine tool, which is 50 Hz. Therefore, the effect of the temperature increase is low and it will not affect the performance of the PSAs and the measurement procedures.

In the designed system, each PSA is used in a preloaded configuration. Figure 5 presents the design of the preloaded PSA configuration as well as the preloading mechanism. Using the PSA in a preloaded configuration can compensate for the inertial forces that exist during the performance of the system. In the designed configuration, a high-bandwidth mechanical fixture guides a nanopositioning mechanism using the preloaded piezoelectric stack actuator. The PSA used in the design is compact, stiff, and multi-layered. The PSAs used in the designed system have a voltage range of 0 to 45 volts, a maximum displacement of 2.0 μm at 45 volts, a hysteresis less than 15%, and a maximum load of 65 N at 45 volts. The resonant frequency of the PSA is 540 KHz at no load, with an impedance of 400 mΩ at the resonant frequency.

#### 3.1.2. Strain Gauge Characteristics

In ultra-high-precision SPDT applications, a strain gauge can be bonded on the surface of the tool holder to measure the change in the resistance of the strain gauge caused by the change in the tool holder surface strain. The magnitude of the mechanical deformation of the tool holder is very small; therefore, the corresponding strain can be expressed as a micro-strain. The strain and micro-strain of the piezoelectric element can be expressed by the following equations:(14)$$\varepsilon i=\frac{∆Li}{Li}$$
(15)$$\mu \varepsilon i=\frac{∆Li}{Li}*10^{-6}$$
where *εi* is the strain, *μεi* the micro-strain, and ∆*L*/*L* is the change in surface strain in the desired dimension. In this condition, the change in the resistance of the strain gauge (gauge factor) can be calculated using the following equation:(16)$${GF}_{\varepsilon_{i}}=\frac{∆R_{i}}{R_{i}}/\frac{∆L_{i}}{L_{i}}$$
where *Ri* is the resistance of the strain gauge, ∆*R* is the change in the resistance of the strain gauge, *G* is the gauge factor and $\varepsilon_{i}$ is the specific strain of the strain gauge component. Therefore, the change in the resistance of the strain gauge can be expressed by the following equation:(17)∆R=R·GF· εi

As illustrated in Figure 2, twelve strain gauges are bonded on the surface of the tool holder, parallel to the *X*-, *Y*-, and *Z*-axes. In this condition, the tool holder body consists of three cantilever beams in which the applied force in the respective direction causes a deflection on the surface and changes the resistance and output voltage of the strain gauge in the respective direction. When an external force is applied to the tool holder body, the tool holder generates a resisting force against the externally applied force (stress) at the same location. In this condition, the tension of the tool holder can be calculated by the following equation:(18)$$\sigma =\frac{MC}{I}$$
where *σ* is the tension of the surface, *M* is the bending moment, *C* is the displacement, and *I* is the moment of inertia. The value of the moment of inertia is in relation to the specifications of the cantilever beam and can be expressed by the following equation:(19)$$I=\frac{bh^{3}}{12}$$
where *I* is the moment of inertia, *b* is the width of the beam, and *h* is the thickness of the beam.

In ultra-high-precision SPDT processes, the cutting speed used is usually in a range of 300–3000 rpm. In this condition, the natural frequency of the machine tool is around 50 Hz. Therefore, the natural frequency of the designed tool holder should be higher than 200 KHz. The natural frequency of the tool holder can be calculated by the following equation:(20)$$f_{n}=\frac{1}{2\lambda }\sqrt{\frac{\frac{3EI}{L^{3}}}{pBDL}}$$
where *f_n_* is the frequency of the tool holder, *E* is the modulus of elasticity of the material, *I* is the moment of inertia, *ρ* is the density of the plate, *B* is its breadth, *D* its width, and *L* is the length. Strain measurement during the diamond-cutting process requires high-precision systems, as the change in the mechanical formation of the tool holder is very small, resulting in small changes in the resistance of the strain gauge. A Wheatstone bridge consists of four arms with resistive elements that can convert the acquired signal to a bipolar voltage output that can be amplified and measured. The output voltage can be measured using the following equation:(21)$$V_{o}=V_{e}*(\frac{R_{3}}{R_{3}+R_{4}}-\frac{R_{2}}{R_{1}+R_{2}})$$
where *V_o_* is the measured output voltage, *V_e_* is the excitation voltage, and *R*_1_–*R*_4_ are the resistor elements. In a Wheatstone configuration, where *R*_1_
*= R*_2_ and *R*_3_
*= R*_4_, when an active strain gauge is used in the configuration of a Wheatstone bridge (*R*_4_ ± ∆*R*), the change in the output of the Wheatstone bridge is in correlation with the change in the resistance of the strain gauge (∆*R*). The Wheatstone bridge allows measuring small changes in the resistance of the strain gauge by using the following equation:(22)$$∆R=R_{s}*GF*\varepsilon i$$
where *Rs* is the resistance of the strain gauge element in no load condition, *GF* is the gauge factor of the strain gauge, and *εi* is the strain factor of the strain gauge. In this condition, by measuring the output voltage of the Wheatstone bridge, the change in the strain can be calculated. The output voltage of the designed bridge network can be expressed by the following equation:(23)$$V_{o}=V_{e}*\left(-GF*\frac{\varepsilon }{4}*\frac{1}{\left(1+(GF*\frac{\varepsilon }{2}\right)}\right)$$

In this condition, the variation of the resistance of the active gauges can change the output voltage. By using the Equation (24), the strain can be measured, and the applied force can be determined:(24)$$V_{o}=V_{e}*-GF*\varepsilon$$

In addition, the sensitivity of the designed bridge can be expressed as:(25)$$Sensitivity=\frac{V_{o}}{V_{e}}=-GF*\varepsilon \;[\frac{mV}{V}]$$

### 3.2. Structure Design of the Smart Cutting Tool

The schematic design of the structure of the smart cutting tool is presented in Figure 6. The mechanical structure of the tool holder body should be rigid and stable during the machining process. It should also be sensitive to small forces applied to the cutting tool during the cutting process. The structure design should also provide a three-degree-of-freedom mechanism for the nanopositioning of the cutting tool while minimizing the possibility of surface strain under the applied forces in SPDT. For this purpose, tool steel material is considered as the substrate material and the specifications of the mechanical design are adopted to the desired conditions.

### 3.3. Control System

The control system should be able to distinguish the small changes in the output voltage of the strain gauges and the PSAs so that the tool position as well as the applied forces can be measured and set. In the designed mechanism, twelve strain gauges and three PSA are used to enable in-process force measurements and tool position control during diamond cutting, while employing three independent control mechanisms, as described below. By using the strain gauges bonded on three surfaces of the tool holder body, the deformation on the surface can be measured and the applied force corresponding to the surface deflection can be indirectly calculated. Moreover, by using the strain gauges bonded on the surface of the tool holder, the small surface deflections (position errors) can be monitored and corrected in real time. The outputs of the strain gauges are used as position feedbacks to identify and correct the position errors that occur. The position feedbacks are used in the designed closed-loop control system to measure the surface deformities caused by the applied force and accurately setting the position of the diamond tool in the desired location. According to the direction of the deformity that occurs on the tool shank surface, which can be determined using the output signal of the strain gauges, the control system increase/decrease the amplitude of the voltage applied to the piezoceramic chips while using the position feedback to achieve the initial (correct) position of the surface and compensate for the positioning displacement of the diamond tool. Moreover, the control system should be able to connect to a hybrid controller and transmit the acquired data when the device is used in a hybrid or intelligent SPDT platform.

Figure 7 presents the detailed design of the control system of the smart cutting tool. In the design of the control system, an Arduino UNO WIFI Rev. 2 development board is used. This controller board has the Microchip ATmega4809 microcontroller, with 14 digital and 6 analog input/output ports, a clock speed of 16 MHz, and integrated WIFI/Bluetooth modules. Three HX711 load cell amplifiers, with 24-bit analog-to-digital converters, are used in the design of the control system to amplify the acquired voltage signals with high precision and to easily read the change in the voltage of the Wheatstone bridges. The HX711 can communicate with the microcontroller via a two-wire interface (data and clock). The Wheatstone bridges are connected to the HX711 amplifiers and the output of the HX711 is connected to digital pins of the microcontroller. The change in the measured voltage can be distinguished via the microcontroller and the corresponding strain of the surface can be calculated. By determining the amount of strain/tension of the tool holder surface, the change in the initial position of the diamond tool can be determined and corrected with high precision.

Three preloaded PSAs are used in the proposed system. When an external force is applied to the diamond tool at the tool nose, the tool holder bends, and the applied force is transferred to the PSAs placed at the other side of the tool holder body. The applied force presses the piezoceramic component at a nanometric scale, which causes a change in the physical properties of the piezoceramic component and changes the resistance and capacitance of the component. The applied force causes the generation of a small amount of electrical charge that can be positive or negative. This causes a change in the drive voltage of the piezoceramic element. By measuring the change in the voltage at the driving/detecting electrodes, the generated electrical charge as well as the applied force can be calculated. By changing the amount of the applied voltage, it is possible to change the position of the cutting tool, as illustrated in Figure 2.

An AC voltage input is connected to two DC–DC transformers to provide the required power to drive the control system and the piezoceramics. The outputs of the transformers are connected to a voltage regulator in which a noiseless input voltage without a voltage drop is secured. By using an LM7808 voltage regulator, a regulated 8 VDC is connected to the control system. In addition, a 0–40 V DC–DC step-down power module using LM2596 is implemented within the system to supply the required voltage for loading the piezoceramic components. A dual H-bridge L298N driver is used to supply the required voltage to charge the piezoceramic components.

By using wireless communication protocols, the control system can connect to a Bluetooth or WIFI network and communicate with a hybrid controller. The control system can transmit data, including the measured cutting forces as well as the position of the diamond tool compared to its initial position, to a network, software applications, and intelligent systems of the hybrid SPDT platform. The control system can also receive commands from the user or hybrid controller to change the position of the diamond tool with a precision down to 1 nanometer and set/correct the positioning errors of the diamond tool during the diamond-cutting process.

The block diagram of the force control and position control loops implemented within the control system are illustrated in Figure 8. The control system has three independent closed-loop control algorithms working simultaneously to measure the applied forces in the *X*-, *Y*-, and *Z*-directions and to control the positioning of the tool. Feedback from the applied force in the respective direction and feedback from the position displacement in respective axis are used within each control subsystem to precisely measure the displacements of the cutting tool and the applied forces. The control system can compensate for the displacements of the cutting tool along the *X*-, *Y*-, and *Z*-axes with nanometer accuracy, by precisely increasing/decreasing the applied voltage to the respective PSA at a micro- to millivolt scale.

#### 3.3.1. The Arrangement Strategy of the Strain Gauges and Piezoelectric Stack Actuators

As illustrated in the Figure 9A, three groups of strain gauges are bonded on the tool holder surface in the area of the highest strain on the surface of tool holder while considering the maximizing the sensitivity of the force measurement along the *X*-, *Y*-, and *Z*-axes. Four active strain gauges *R*_1_, *R*_2_, *R*_3_, and *R*_4_ are arranged on the surface of the tool holder along the *X*-axis to detect the back force and tool holder surface strain along the *X*-axis. Another four active strain gauges *R*_5_, *R*_6_, *R*_7_, and *R*_8_ are arranged on the surface of the tool holder along the *Y*-axis to detect the cutting force and tool holder surface strain along the *Y*-axis. Moreover, four active strain gauges *R*_9_, *R*_10_, *R*_11_, and *R*_12_ are arranged on the surface of the tool holder along the *Z*-axis to detect the feed force and tool holder surface strain along the *Z*-axis. The strain gauges are bonded on the surfaces in Wheatstone bridge configurations.

The resistance of the strain gauges vary when the external forces are applied to the tool holder structure. Table 1 presents the variations in the resistance of strain gauges bonded on the tool holder. *F_x_* indicates the back force applied along the *X*-axis, *F_y_* indicates the cutting force applied along the *Y*-axis, and *F_z_* indicates the applied feed force along the *Z*-axis, as illustrated in Figure 2. As illustrated in Figure 9A, the excitations of the Wheatstone bridges are connected to voltage excitation, and the corresponding output voltage ∆*V_i_* appear on the measurement terminals of the bridges. The following equations express the output signals of the Wheatstone bridges:(26)$${∆V}_{x}=\frac{V_{e}}{4}\left(\frac{∆R_{1}}{R_{1}}-\frac{∆R_{2}}{R_{2}}-\frac{∆R_{3}}{R_{3}}-\frac{∆R_{4}}{R_{4}}\right)$$
(27)$${∆V}_{y}=\frac{V_{e}}{4}\left(\frac{∆R_{5}}{R_{5}}-\frac{∆R_{6}}{R_{6}}-\frac{∆R_{7}}{R_{7}}-\frac{∆R_{8}}{R_{8}}\right)$$
(28)$${∆V}_{z}=\frac{V_{e}}{4}\left(\frac{∆R_{9}}{R_{9}}-\frac{∆R_{10}}{R_{10}}-\frac{∆R_{11}}{R_{11}}-\frac{∆R_{12}}{R_{12}}\right)$$
where ∆*R_i_/R_i_* represents the resistance variation rate of the gauge *R_i_*, ∆*V_x_* is the change in the output voltage of the Wheatstone bridge along the *X*-axis, ∆*V_y_* is the change in the output voltage of the Wheatstone bridge along the *Y*-axis, ∆*V_z_* is the change in the output voltage of the Wheatstone bridge along the *Z*-axis, and *R_i_* represents the number of strain gauges in each Wheatstone bridge configuration.

As illustrated in Figure 9B, three PSAs are fixed to the mechanical structure while mechanically interacting with the moving part of the tool holder in the *X*-, *Y*-, and *Z*-directions. The PSA I can push and move the tool holder along the *X*-axis, PSA II can push and move the tool holder along the *Y*-axis, and PSA III can push and move the tool holder along the *Z*-axis. The PSAs are installed in the *g*_33_ polarization direction so that the change in the length of each PSAs can be effective and move the tool holder structure. Additionally, when an external force is applied to the cutting tool, the tool holder body is deflected, and the applied force is transferred to the respective PSA.

#### 3.3.2. Strain Gauge Outputs: Decoupling the Applied Forces and Respective Surface Strain

During the diamond-cutting process, the applied forces, including the cutting force (*F_y_*), back force (*F_x_*), and feed force (*F_z_*), act at the tool nose and deform the tool holder structure in the nanometric scale. Twelve strain gauges are configured in three independent Wheatstone bridges to measure the change in the surface strain corresponding to the applied force along each axis. Three DC output voltages can be acquired, amplified, and used to calculate the applied force corresponding to the change in the output voltage of each Wheatstone bridge. During the measurement, a specific amount of DC voltage is applied to the input terminals of each Wheatstone bridge. The applied force changes the surface strain while the resistance of the strain gauges is changed based on the stress/tension applied to the surface bonded on the tool holder. The relationship between the changes in the output voltage and the applied force can be expressed by the following equation:(29)$$\left\{\begin{array}{c} ∆Vx=K_{sx}\cdot F_{x}\\ ∆Vy=K_{sy}\cdot F_{y}\\ ∆Vz=K_{sz}\cdot F_{z} \end{array}\right.$$where ∆*V*_x_, ∆*V_y_*, and ∆*V_z_* are the respective voltage changes for each Wheatstone bridge under the applied forces *F_x_*, *F_y_*, and *F_z_*, respectively. *K_sx_*, *K_sy_*, and *K_sz_* are corresponding constants to indicate the relationship between the change in the output voltage and the respective force applied. The decoupling measurement algorithm implemented within the control system is expressed as:(30)$$\left[\begin{array}{c} F_{x}\\ F_{y}\\ F_{z} \end{array}\right]=\left[\begin{array}{ccc} {1/K}_{sx} & 0 & 0\\ 0 & 1/K_{Sy} & 0\\ 0 & 0 & 1/K_{sz} \end{array}\right]\left[\begin{array}{c} {∆V}_{x}\\ {∆V}_{y}\\ {∆V}_{z} \end{array}\right]$$

When an external force is applied to the tool holder body, the change in the surface strain on the three faces of the tool holder body is measured. By using Equation (24), the following equation can express the relationship between the applied force and the change in surface strain:(31)$$\left\{\begin{array}{c} {∆\varepsilon }_{x}=K_{\varepsilon x}\frac{∆Vx}{\pm GF\cdot Ve}\\ {∆\varepsilon }_{y}=K_{\varepsilon y}\frac{∆Vy}{\pm GF\cdot Ve}\\ {∆\varepsilon }_{z}=K_{\varepsilon z}\frac{∆Vz}{\pm GF\cdot Ve} \end{array}\right.$$
where ∆εx, ∆εy, and ∆εz are respective changes in the strain of the tool holder surface under the applied forces *F_x_*, *F_y_*, and *F_z_*, respectively. $K_{\varepsilon z}$, $K_{\varepsilon z}$, and $K_{\varepsilon z}$ are corresponding constants to indicate the relationship between the surface strain change and the respective forces applied. ∆Vx, ∆Vy, and ∆Vz are the respective changes in the output voltage of the strain gauge bridge along the *X*-, *Y*-, and *Z*-axes respectively. *V_e_* is the input voltage applied to the strain gauge bridge and *GF* is the gauge factor of the strain gauge bridge. The decoupling measurement algorithm implemented within the control system is expressed as:(32)$$\left[\begin{array}{c} {∆\varepsilon }_{x}\\ {∆\varepsilon }_{y}\\ {∆\varepsilon }_{z} \end{array}\right]=\left[\begin{array}{ccc} \frac{K_{\varepsilon x}}{\pm GF\cdot Ve} & 0 & 0\\ 0 & \frac{K_{\varepsilon y}}{\pm GF\cdot Ve} & 0\\ 0 & 0 & \frac{K_{\varepsilon z}}{\pm GF\cdot Ve} \end{array}\right]\left[\begin{array}{c} {∆V}_{x}\\ {∆V}_{y}\\ {∆V}_{z} \end{array}\right]$$

Therefore, when an external force is applied to the tool holder structure, the surface strain, which causes tool positioning errors in the corresponding axis, can be determined. To correct the displacement, the preloaded PSAs are driven while pushing the tool holder by increasing the applied voltage at the terminals until the surface strain is compensated for and ∆εi = 0.

#### 3.3.3. Piezo Stack Actuators: Decoupling the Applied Force and Respective Surface Strain

When an external force is applied to a piezo stack, it can change the electrical charge and physical properties of the piezoelectric component. Applying a force can generate a negative/positive electrical charge according to the position and direction of the force applied. This can be used as feedback to determine the magnitude of the applied force. The change in the electrical characteristics of the piezoceramic chip can be measured by measuring the input/output voltage at the terminals of each piezoceramic component. By using Equation (13), the following equation expresses the relationship between the applied force and the change in the electrical charge of the PSAs:(33)$$\left\{\begin{array}{c} ∆U_{x}=K_{px}\frac{{g_{33px}\cdot h\cdot F}_{x}}{A}\\ ∆U_{y}=K_{py}\frac{{g_{33py}\cdot h\cdot F}_{y}}{A}\\ ∆U_{z}=K_{pz}\frac{{g_{33pz}\cdot h\cdot F}_{z}}{A} \end{array}\right.$$
where $∆U_{x}$ is the change in the electrical charge of PSA_x_, $∆U_{y}$ is the change in the electrical charge of PSA_y_, and $∆U_{z}$ is the change in the electrical charge of PSA_z_. *Kpx*, *Kpy*, and *K_pz_* are corresponding constants to indicate the relationship between the change in the electrical charge and the respective force applied. *h* is the height of the piezoceramic chip, *g_33_* represents the 33 polarization direction of the piezoelectric material, *A* is the area of each piezoceramic chip, and *F_x_*, *F_y_*, and *F_z_* are the applied forces in the *X*-, *Y*-, and *Z*-directions. By measuring the change in the electrical charge of each PSA, the control system can calculate the corresponding applied force. The decoupling measurement algorithm implemented within the control system can be expressed as:(34)$$\left[\begin{array}{c} F_{x}\\ F_{y}\\ F_{z} \end{array}\right]=\left[\begin{array}{ccc} \frac{A}{K_{px}\cdot g_{33}\cdot h} & 0 & 0\\ 0 & \frac{A}{K_{py}\cdot g_{33}\cdot h} & 0\\ 0 & 0 & \frac{A}{K_{pz}\cdot g_{33}\cdot h} \end{array}\right]\left[\begin{array}{c} {∆U}_{x}\\ {∆U}_{y}\\ {∆U}_{z} \end{array}\right]$$

Moreover, by measuring the change in the electrical potential of the PSA, the change in the height of the PSA can be calculated and the position error in the corresponding axis can be determined. The following equation can express the relationship between the change in the height of the piezoelectric chips and the change in their electrical charge:(35)$$\left\{\begin{array}{c} ∆h_{px}=K_{px}\frac{{g_{33px}\cdot h\cdot F}_{x}}{A}\\ ∆h_{py}=K_{py}\frac{{g_{33py}\cdot h\cdot F}_{y}}{A}\\ ∆h_{pz}=K_{pz}\frac{{g_{33pz}\cdot h\cdot F}_{z}}{A} \end{array}\right.$$
where $∆h_{px}$ is the change in the height of PSA_x_, $∆h_{py}$ is the change in the height of PSA_y_, and $∆h_{pz}$ is the change in the height of PSA_z_. *Kpx*, *Kpy*, and *K_pz_* are corresponding constants to indicate the relationship between the change in the electrical charge and the respective force applied. *h* is the height of the PSA, *g*_33_ represents the 33 polarizations direction of the PSA, *A* is the area of each PSA, and *F_x_*, *F_y_*, and *F_z_* are applied forces in the *X*-, *Y*-, and *Z*-directions. By measuring the change in the electrical potential of each PSA, the change in the height of the chip corresponding to position displacement in respective axis can be calculated. The decoupling measurement algorithm implemented within the control system can be expressed as:(36)$$\left[\begin{array}{c} {∆h}_{x}\\ {∆h}_{y}\\ {∆h}_{z} \end{array}\right]=\left[\begin{array}{ccc} \frac{A}{K_{px}\cdot g_{33}\cdot F_{x}} & 0 & 0\\ 0 & \frac{A}{K_{py}\cdot g_{33}\cdot F_{y}} & 0\\ 0 & 0 & \frac{A}{K_{pz}\cdot g_{33}\cdot F_{z}} \end{array}\right]\left[\begin{array}{c} {∆U}_{x}\\ {∆U}_{y}\\ {∆U}_{z} \end{array}\right]$$

### 3.4. Finite Element Analysis (FEA) Validation

To verify the designed mechanism and the performance of the proposed smart cutting tool in terms of surface strain changes under the applied cutting forces, a finite element analysis (FEA) was performed using Autodesk Inventor Professional 2021 software. The specifications of the designed mechanical structure are presented in Table 2 and its architectural design is illustrated in Figure 10.

Regarding the tool holder material, tool steel is an ideal choice for tool holder applications. Tool steel has enhanced mechanical properties compared to other materials and can improve the cutting tool performance [45]. Tool steel grade O1 is considered as the substrate material for the manufacturing of the tool holder due to its distinctive hardness, resistance to abrasion and deformation, and ability to hold a cutting edge at elevated temperatures. The details of this material are: AISI grade = O1; chemical composition: 0.90% C, 1.0–1.4% Mn, 0.50% Cr, 0.50% W, 0.30% Si, and 0.20% V; hardness = 61HCR; Young’s modulus-E = 214 GPa [46].

The stiffness is an important index that can be defined by dividing the input load by the corresponding displacement of the input end of the mechanism. The effect of applying an external force on the surface strain is simulated using six models to indicate the effect of the back force, cutting force, feed force, PSA_x_ force, PSA_y_ force, and PSA_z_ force applied to the tool holder in predetermined directions/positions. The results of surface strain analysis including the applied force and the corresponding deformation displacements are presented in Figure 11, Figure 12, Figure 13, Figure 14, Figure 15 and Figure 16. A back force of 5 N, a cutting force of 5 N, a feed force of 5 N, a pushing force of 5 N for PSA_x_, a pushing force of 5 N for PSA_y_, and a pushing force of 5 N for PSA_z_ are set and used in the modeling and analysis of the designed smart cutting tool.

Taking into account that a displacement at the nanometric scale occurs in the diamond-cutting process, a displacement of 0.1 μm is exerted on the input end in each condition, and the displacement and amplification ratio are obtained. An FEA static stress analysis was also performed to assess the performance and safety of the designed platform by applying the respective input force/displacements at the input end of the designed mechanism. The results of the analysis, including the range of stress applied to the surface and the maximal stress are presented in Figure 11, Figure 12, Figure 13, Figure 14, Figure 15 and Figure 16. In the stress analysis, single-point mode with fixed boundary conditions and an element size of 1.5 are set. It can be deduced from the results that in all test conditions, the designed platform has high strength, with a safety factor of around 15 uL which guarantees good linearity, precision, and repeatability during the measurement and tool position control process.

## 4. Discussion

In ultra-precision manufacturing, the application of in-process cutting force measurements is well established. In ultra-high-precision SPDT technology, the cutting force is a critical factor and directly influences the outcome of the machining process in terms of the quality of the surface finish. Due to recent developments on piezoelectric materials and strain gauges, different force measurement techniques have been developed to measure the interactive forces, including the back force, cutting force, and feed force, in the ultra-precision diamond-cutting process. On the other hand, with the development of advanced PSAs, the application of nanopositioning systems in ultra-precision machining applications has emerged. Nanopositioning systems can be used to enable tool servo systems, and to precisely set the position of the cutting tool while enabling the compensation for the position errors of the cutting tool in ultra-precision machining applications.

In this research work, by combining a three-dimensional surface strain measurement system and a three-dimensional nanopositioning system, a hybrid cutting tool was developed to be used in ultra-high-precision SPDT applications. The proposed smart cutting tool can measure the applied forces to the diamond tool in the diamond-cutting process in three dimensions. Two types of position feedbacks are used to determine the displacement of the diamond tool along the *X*-, *Y*-, and *Z*-axes. Three strain gauge measurement systems are bonded on three surfaces of the tool holder to measure the surface deflections in three dimensions. Moreover, three PSAs are fixed to different surfaces of the moving part of the tool holder and mechanically interacted with the tool holder for the nanopositioning of the cutting tool. In the designed configuration, the cutting force (*F_y_*), feed force (*F_z_*), and back forces (*F_x_*) are transferred to piezoceramic chips while changing the electrical charge of the chips and compressing them at nanometric scales. The applied force to each PSA can be calculated using the change in the electrical charge of the chip. Subsequently, the cutting force, feed force, and back forces applied to the diamond tool can be determined using the change in the surface strain as well as the change in the length of the respective PSA.

Moreover, by implementing three PSAs, a high-precision 3 DOF XYZ nanopositioning system with nanometric resolution is enabled. The designed control system can compensate for the surface deflections and displacement of the diamond tool by applying small changes to the drive voltage of each PSA and change the length of each PSA by increasing/decreasing the applied drive voltage. By increasing/decreasing the amount of the drive voltage applied to the piezoceramic chip, the length of the PSA can be precisely controlled. Therefore, it is possible to compensate for the displacement of the tool holder and the diamond tool by applying a precise voltage level that resets the position of the diamond tool to its initial position with nanometer positioning accuracy. The output of the strain gauge systems as well as the change in the drive voltage level of each PSA are the feedback used for compensating the position errors in three dimensions.

By using the transverse piezoelectric effect as well as the symmetric arrangement of the sensors integrated with the delicate design of the tool holder structure, the relationships between the applied forces and respective electric charges are established. Subsequently, the decoupling algorithms are derived. The FEA on surface stress of the tool holder was performed to simulate and analyze the performance of the tool holder under six conditions to investigate the effects of the applied forces on the surface deflection of the tool holder. The FEA results show that the designed structure is suitable to be used in ultra-high-precision SPDT applications. The simulation results are shown to be almost consistent with the results of the derived analytical model and justify the functionality of the designed tool holder for in-process force measurement applications and nanopositioning of the diamond tool in the ultra-precision SPDT process. In the diamond-cutting process, small changes in the magnitude of the cutting force impact the machining conditions and the outcome of the process, which can be monitored for process optimizations. For the measurement of the cutting force during the SPDT process, high-precision systems with a sampling accuracy better than 0.1 N are required [13,24]. According to the design specifications, the proposed smart cutting tool has a bandwidth of 5 N and a high sensitivity and sampling accuracy. The designed system has high performance and can measure the micro-cutting force in the *X*-, *Y*-, and *Z*-directions with high precision.

## 5. Conclusions

A novel smart cutting tool is designed and analyzed for in-process cutting force measurement and tool position control in ultra-high-precision SPDT applications. The proposed device is a hybrid tooling system that can measure the applied forces to the diamond tool in three dimensions. It can also measure the deflection of the tool holder surface as well as the displacement of the diamond tool under applied forces during the SPDT process. By using a high-precision nanopositioning system using PSAs, tool position can be controlled at the nanometer scale and the unwanted displacements of the diamond tool, caused by tool holder strain/tension under the cutting force, can be compensated for in real time. The proposed system has a high-precision and can be used in ultra-high-precision SPDT applications. In a hybrid SPDT platform, the proposed system can be used as part of the hybrid/intelligent machining platform with the capability of communicating with the hybrid controller using wireless communication protocols.

A prototype is expected to be fabricated to experimentally investigate the dynamic, kinematic, and error analysis of the proposed system in upcoming work. Future work will focus on the optimization of the dimension parameters of the smart cutting tool to achieve optimum performance. The proposed smart cutting tool is designed to sense interactive forces during the diamond-cutting process as well as for nanopositioning of the diamond tool in ultra-high-precision SPDT applications, where cutting forces are at 0.1 N and tool displacements are at 1 nm scales.

## 6. Patents

A provisional patent application for the Netherlands was produced from the research work reported in this manuscript. The proposed smart cutting tool, including the method of force measurement and the nanopositioning system, is under the protection of Netherlands Intellectual Property Office, as an invention patent, with application number N2035244, entitled: “Ultra-precision smart cutting tool for in-process force measurement and tool nanopositioning”.

## Figures and Tables

**Figure 1 micromachines-14-01857-f001:**
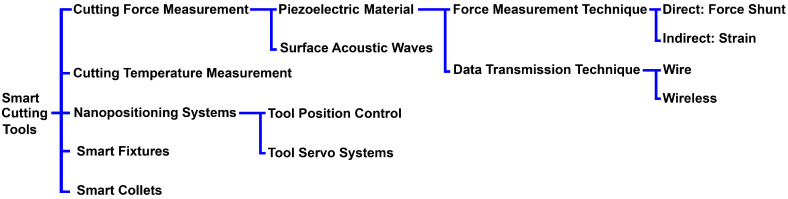
Classification of smart cutting tools in ultra-precision SPDT.

**Figure 2 micromachines-14-01857-f002:**
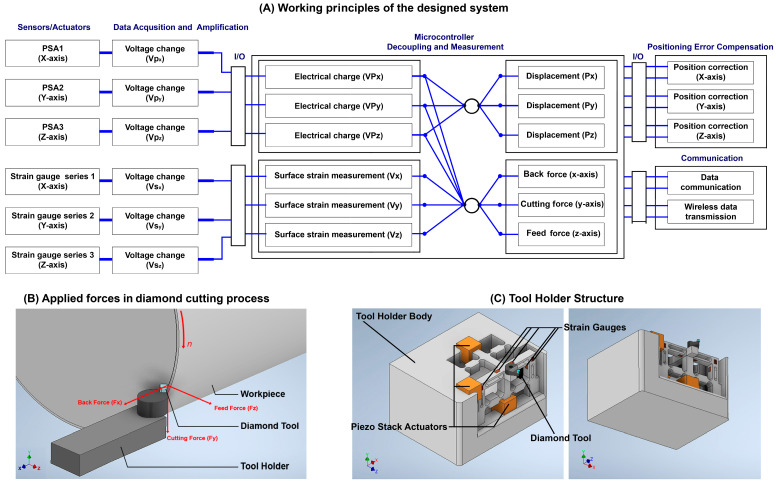
(**A**) Working principles of the designed smart cutting tool. (**B**) Illustration of applied forces during single-point diamond-turning process. (**C**) Schematic design of the smart cutting tool.

**Figure 3 micromachines-14-01857-f003:**
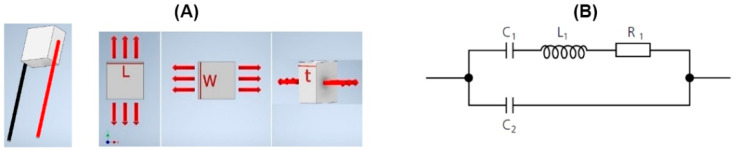
(**A**) Modes of displacement/vibration in rectangular plate piezoelectric component; L: length; W: width; t: thickness. (**B**) Equivalent circuit diagram of piezoelectric component; C1: inverse stiffness; C2: dielectric capacity; L1: inter mass; R1: interior loss.

**Figure 4 micromachines-14-01857-f004:**
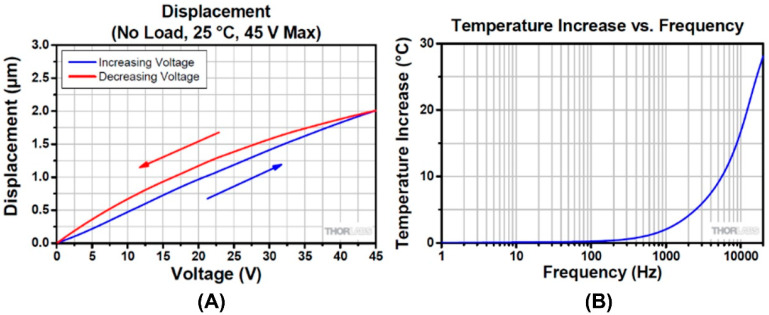
The performance of a typical piezo stack actuator: (**A**) The displacement vs. voltage hysteresis graph (H < 15%). (**B**) Frequency vs. temperature increase graph [44].

**Figure 5 micromachines-14-01857-f005:**
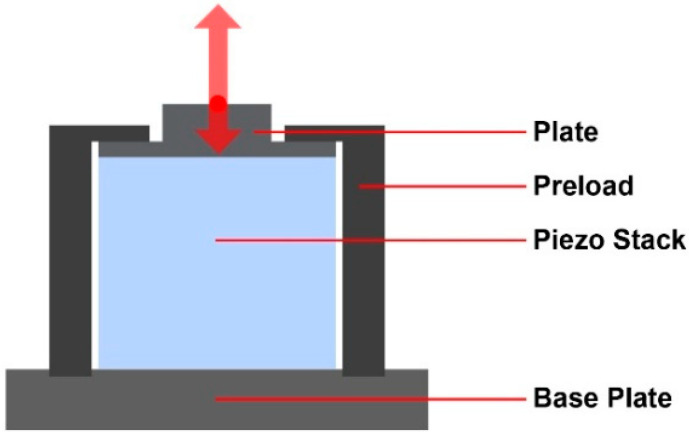
The preloading mechanism of the piezo stack actuator.

**Figure 6 micromachines-14-01857-f006:**
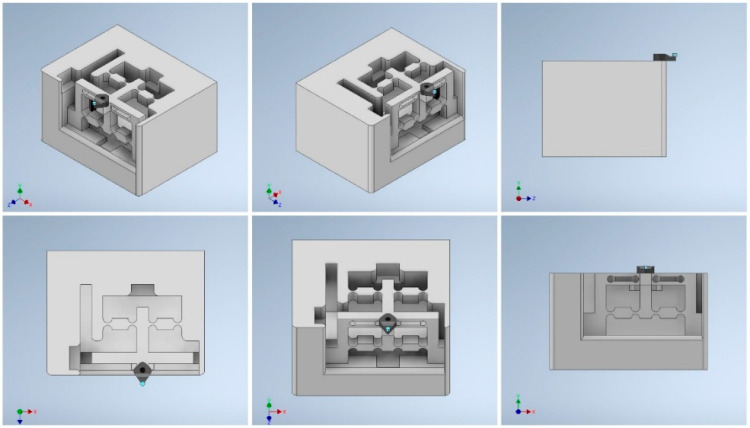
The structure design of the tool holder body.

**Figure 7 micromachines-14-01857-f007:**
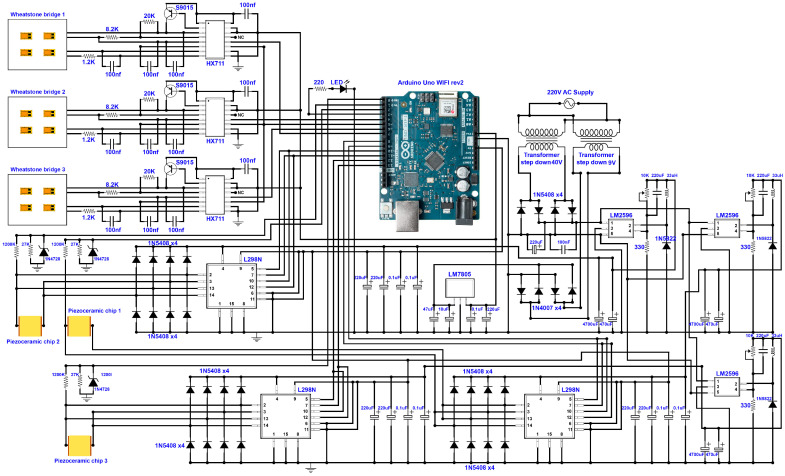
Design of the control system.

**Figure 8 micromachines-14-01857-f008:**
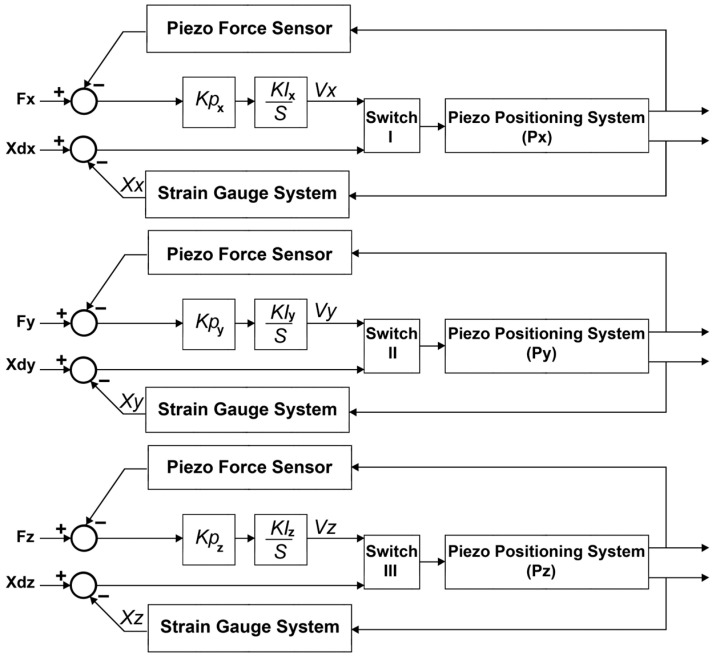
Block diagram of the designed closed-loop control system.

**Figure 9 micromachines-14-01857-f009:**
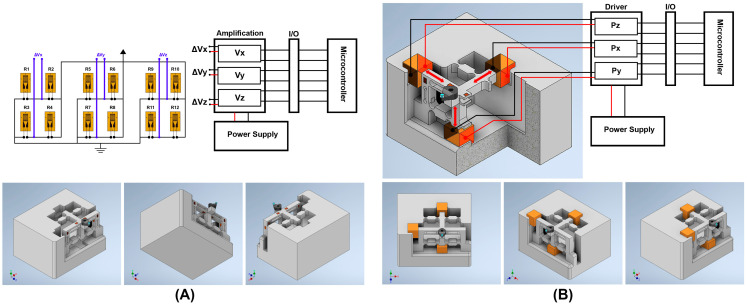
The arrangement strategy of the sensors/actuators: (**A**) The electrical configuration and arrangement of the strain gauges; (**B**) The electrical configuration and arrangement of piezo stack actuators.

**Figure 10 micromachines-14-01857-f010:**
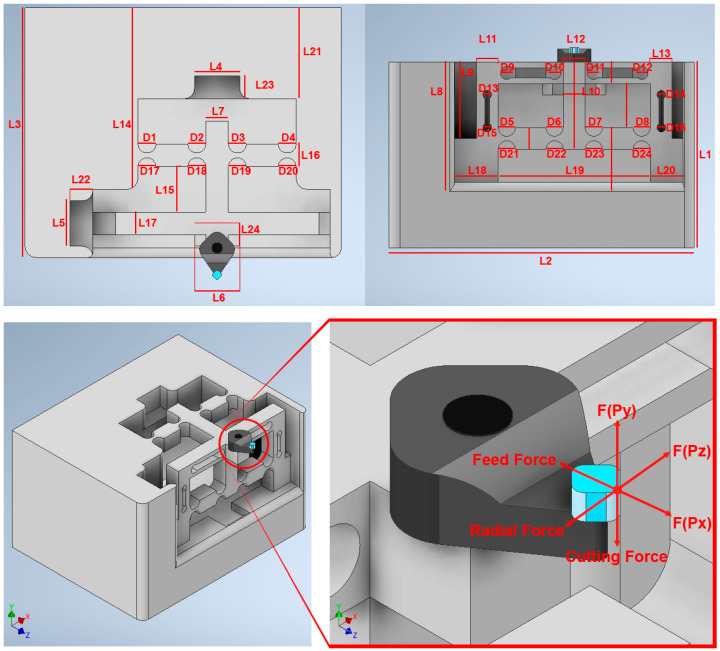
Architectural design of the smart cutting tool and the reachable workspace of the diamond tool under interactive forces.

**Figure 11 micromachines-14-01857-f011:**
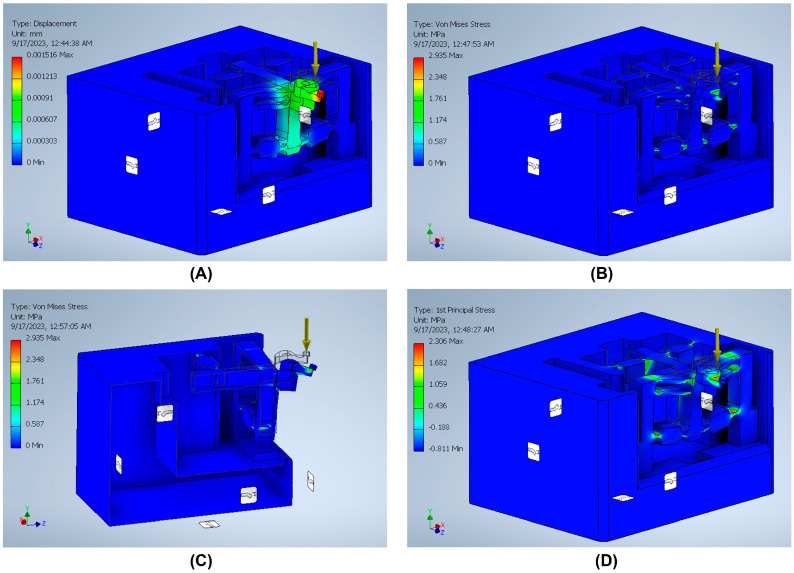
Static stress analysis of the tool holder under the applied force (*Fy*): (**A**) displacement; (**B**) Von Mises stress; (**C**) sectional view of the Von Mises stress; (**D**) 1st principal stress.

**Figure 12 micromachines-14-01857-f012:**
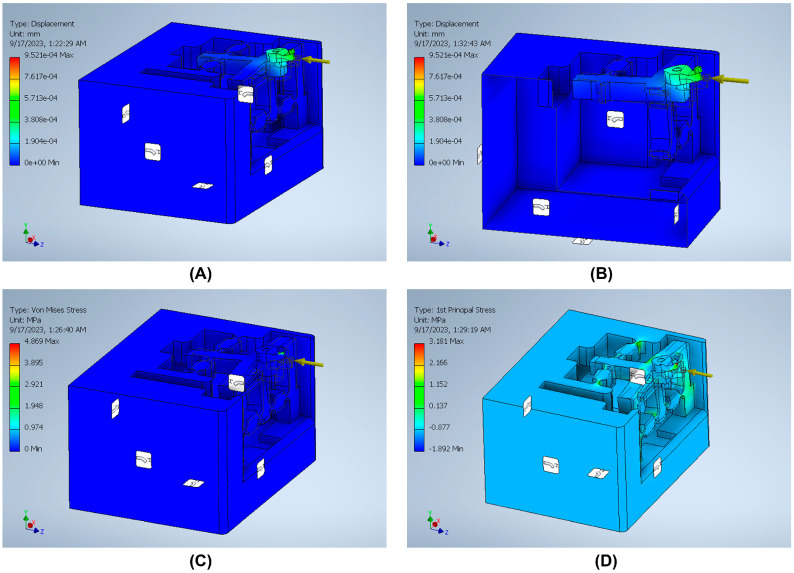
Static stress analysis of the tool holder under the applied force (*Fx*): (**A**) displacement; (**B**) sectional view of the displacement; (**C**) Von Mises stress; (**D**) 1st principal stress.

**Figure 13 micromachines-14-01857-f013:**
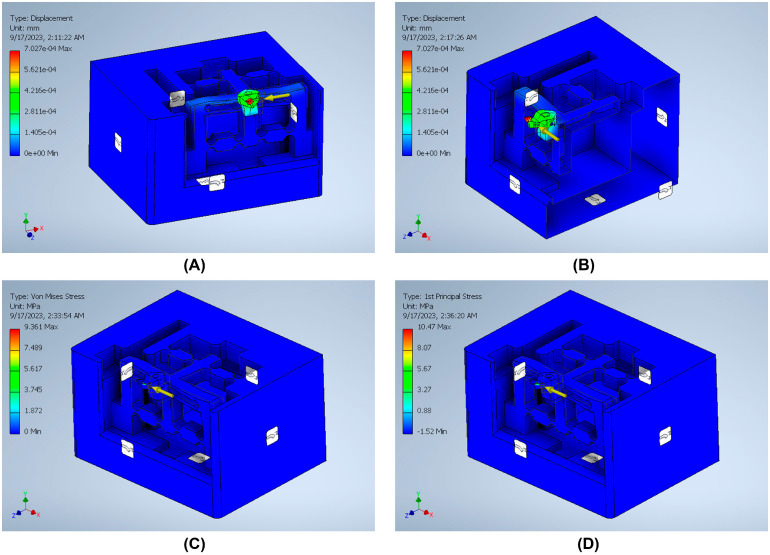
Static stress analysis of the tool holder under the applied force (*Fz*): (**A**) displacement; (**B**) sectional view of the displacement; (**C**) Von Mises stress; (**D**) 1st principal stress.

**Figure 14 micromachines-14-01857-f014:**
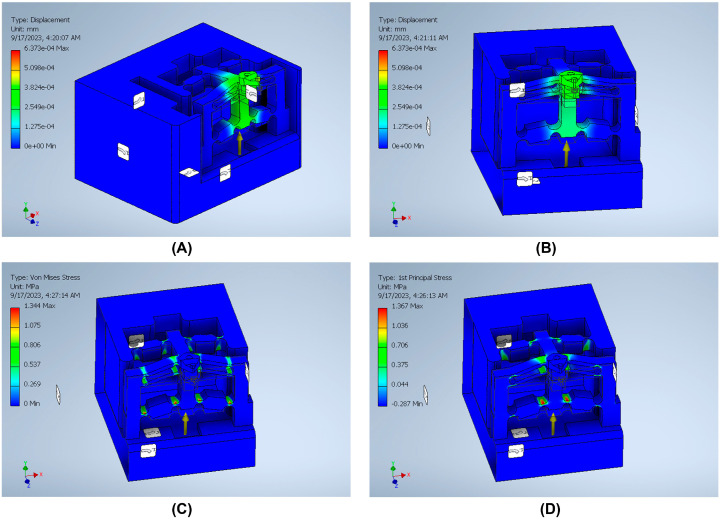
Static stress analysis of the tool holder under the applied force by PSA_y_: (**A**) displacement; (**B**) sectional view of the displacement; (**C**) sectional view of the Von Mises stress; (**D**) sectional view of the 1st principal stress.

**Figure 15 micromachines-14-01857-f015:**
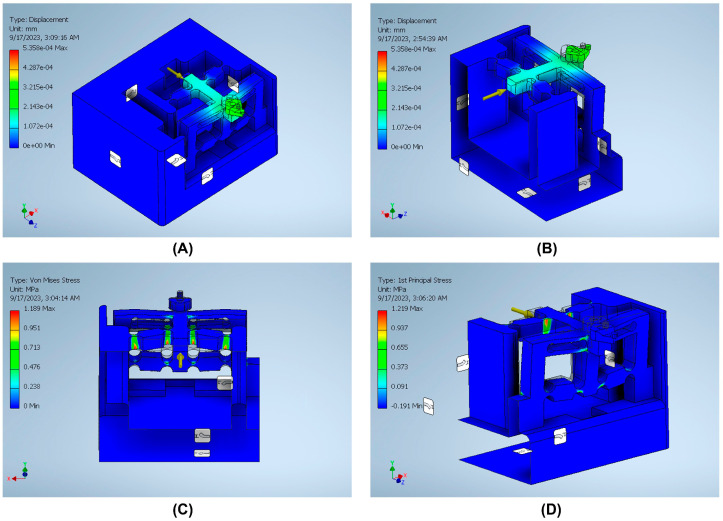
Static stress analysis of the tool holder under the applied force by PSA_x_: (**A**) displacement; (**B**) sectional view of the displacement; (**C**) sectional view of the Von Mises stress; (**D**) sectional view of the 1st principal stress.

**Figure 16 micromachines-14-01857-f016:**
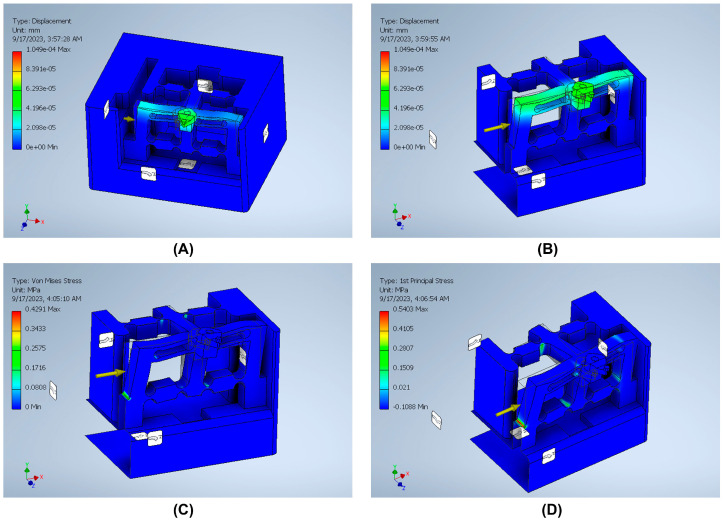
Static stress analysis of the tool holder under the applied force by PSA_z_: (**A**) displacement; (**B**) sectional view of the displacement; (**C**) sectional view of the Von Mises stress; (**D**) sectional view of the 1st principal stress.

**Table 1 micromachines-14-01857-t001:** The variation of resistance of the strain gauges.

Measurement System	Strain Gauge	Back Force (*F_x_*)	Cutting Force (*F_y_*)	Feed Force (*F_z_*)
Wheatstone bridge I	*R* _1_	+	0	0
	*R* _2_	−	0	0
	*R* _3_	+	0	0
	*R* _4_	−	0	0
Wheatstone bridge II	*R* _5_	0	+	0
	*R* _6_	0	−	0
	*R* _7_	0	+	0
	*R* _8_	0	−	0
Wheatstone bridge III	*R* _9_	0	0	+
	*R* _10_	0	0	−
	*R* _11_	0	0	+
	*R* _12_	0	0	−

**Table 2 micromachines-14-01857-t002:** Architectural parameters of the designed smart cutting tool.

Symbol	L1	L2	L3	L4	L5	L6	L7	L8	L9	L10	L11	L12
Value (mm)	85	140	100	20	20	20	10	60	40	10	10	10
Symbol	L13	L14	L15	L16	L17	L18	L19	L20	L21	L22	L23	L24
Value (mm)	10	80	20	10	10	20	70	15	40	10	10	10
Symbol	D1	D2	D3	D4	D5	D6	D7	D8	D9	D10	D11	D12
Value (mm)	8	8	8	8	8	8	8	8	6	6	6	6
Symbol	D13	D14	D15	D16	D17	D18	D19	D20	D21	D22	D23	D24
Value (mm)	4	4	4	4	8	8	8	8	8	8	8	8

## Data Availability

The research data related to this work are included within the manuscript. For more information on the data, contact the corresponding authors.

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
