# Peer review of "Design and Analysis of Ultra-Precision Smart Cutting Tool for In-Process Force Measurement and Tool Nanopositioning in Ultra-High-Precision Single-Point Diamond Turning"

_micromachines, 2023, doi:10.3390/mi14101857_

Round 1
Reviewer 1 Report
The paper presents a practical smart cutting tool for in-process force measurement, and it can be used in SPDT platforms. The work has industrial relevance, is innovative, and has a good application prospect. The following factors need to be addressed for publication:
1. The abstract section's background is overwhelming and requires modification.
2. In the Introduction section, some scholars' research results are cited, but there is a lack of detailed data. Specific data need to be given for comparison with this paper.
3. Please list the performance characteristics of the PSZ used in the manuscript.
4. In the attached pictures of Table 2, some of the size labels are unclear due to color distortion. Please edit them for clarity.
5. In Section 3.3, It is recommended to list the BOM.
6 . In Section 3.4, Please give the mesh, and boundary condition information used.
7. In Figures 11-16, the stress data analyzed is not clearly expressed. Please use a partial view to show the stress data.
8. The measurement data may be affected by the screw connection and contact with the holder and should be analyzed accordingly.
Reviewer 2 Report
The results of the paper are reasonable and acceptable. I suggest minor revision.
1. How have the authors contributed differently from other research groups?
2. Figure 4 (B) lacks a description in the text.
3. Figures 9, 11, 12, 13, 14, 15, and 16 require enhancements. The text within these figures is too small for legibility in the printed version.
4. Piezo hysteresis is observed in Figure 4 (A). How did the authors address or correct this phenomenon?
